# Mechanism of the Pulvinus-Driven Leaf Movement: An Overview

**DOI:** 10.3390/ijms25094582

**Published:** 2024-04-23

**Authors:** Fanwei Zeng, Zonghuan Ma, Yongqing Feng, Miao Shao, Yanmei Li, Han Wang, Shangwen Yang, Juan Mao, Baihong Chen

**Affiliations:** College of Horticulture, Gansu Agricultural University, Lanzhou 730070, China; zengfanw@yeah.net (F.Z.); mazohu@163.com (Z.M.); 18893425315@163.com (Y.F.); 15842120776@163.com (M.S.); l18309467949@163.com (Y.L.); wangh199749@163.com (H.W.); 15596765903@163.com (S.Y.); maojuan@gsau.edu.cn (J.M.)

**Keywords:** aquaporins, actin, ions transport, leaf movement, pulvinus, water transport

## Abstract

Leaf movement is a manifestation of plant response to the changing internal and external environment, aiming to optimize plant growth and development. Leaf movement is usually driven by a specialized motor organ, the pulvinus, and this movement is associated with different changes in volume and expansion on the two sides of the pulvinus. Blue light, auxin, GA, H^+^-ATPase, K^+^, Cl^−^, Ca^2+^, actin, and aquaporin collectively influence the changes in water flux in the tissue of the extensor and flexor of the pulvinus to establish a turgor pressure difference, thereby controlling leaf movement. However, how these factors regulate the multicellular motility of the pulvinus tissues in a species remains obscure. In addition, model plants such as *Medicago truncatula*, *Mimosa pudica*, and *Samanea saman* have been used to study pulvinus-driven leaf movement, showing a similarity in their pulvinus movement mechanisms. In this review, we summarize past research findings from the three model plants, and using *Medicago truncatula* as an example, suggest that genes regulating pulvinus movement are also involved in regulating plant growth and development. We also propose a model in which the variation of ion flux and water flux are critical steps to pulvinus movement and highlight questions for future research.

## 1. Introduction

Movement is an important basic physiological characteristic in the plant kingdom, and it is evolved by plants to adapt to the changing environment and regulate their growth and development [1]. For example, leaf movement, phototropic movement, circadian movement, and stomatal movement in plants are driven by changes in turgor pressure in certain tissues [1,2,3]. Leaf movement is a typical movement regulated by the pulvinus motor organ [4]. Ever since the great botanist Charles Robert Darwin published his book “The Power of Movement in Plants” in 1880, scientists from various disciplines have become very interested in the movement of plants, triggering a research boom that has lasted to this day [5]. Over the years, scientists have adopted numerous techniques of study depending on pulvinus-driven leaf movements, such as patch-clamp techniques, isotope tracers, nuclear magnetic resonance, scanning electron microscopy, flame photometry, and pharmacology experiments [6].

At present, there are numerous model plants for studying pulvinus-driven leaf movement, such as *Medicago truncatula* (*M. truncatula*) (Figure 1A,B), *Mimosa pudica* (*M. pudica*) (Figure 1C,D), and *Samanea saman* (*S. saman*) (Figure 1E,F) [3,7,8]. According to the relationship between movement direction and external stimulus, the movement of the above plants is classified as nastic movement, which refers to the local movement of the plant body caused by undirected external stimuli [9]. Importantly, these plants have a well-developed motor organ (Figure 1G–I) called the pulvinus, also known as the osmotic motor [10,11]. The morphological structure changes from a liner structure to a curved structure after the pulvinus organ drives the leaf movement, and it is divided into the extensor side and the flexor side [12,13] (Figure 2).

Light can be used both as a source of energy and a signal factor for the change in plant morphology and structure [7,14]. The research showed that blue-light (BL)-induced single-leaf movement of *S. saman* by inducing the release of K^+^ on the flexor side of the pulvinus [7] (Figure 3). Furthermore, the characterization of K^+^ channel function by a patch-clamp technique under different light conditions confirmed that K^+^ efflux, a determining factor for the cell-volume decrease of flexor cells, is regulated by BL in a dual manner via membrane potential and by an independent signaling pathway [7]. Phytohormones coordinate the overall growth and development of plants and also participate in plant movement physiological activities [15]. Application of IAA solutions at concentrations greater than 30 µg/mL induced the opening of closed mimosa leaflets even in darkness [16]. Protoplasts isolated from the pulvinus and bathed in a medium containing KCl as the major salt were found to swell in response to IAA and to shrink in response to ABA [17].

In addition, changes in tissue anion and cation flow signals, the osmotic potential generated by the differential distribution of various ions such as K^+^, Cl^−^, and Ca^2+^ in the tissue, and the water transport regulated by aquaporins are believed to regulate the volume of moving cells in the pulvinus, thus determining leaf movement [8,18,19]. Light, phytohormones, ions, and aquaporins are characterized by a complex network connected together [20,21,22] (Figure 3). However, previous studies have only characterized how a single factor affects the pulvinus movement and have not clearly analyzed how these factors interact. Excavating the key genes regulating the pulvinus movement from the transcriptional level is the focus of future research. Notably, studies have found that leaf-movement traits of *M. truncatula* are closely related to a variety of agricultural traits through gene-editing technology, and leaf-movement loss caused by the shortening of pulvinus structure leads to the reduction of plant branching, plant dwarfing, and the occurrence of leaf morphology [3,23,24,25,26]. This provides many unique perspectives for the study of leaf movement, and it becomes vital to reveal the mechanism of leaf movement. However, how light, phytohormones, and ions work together to regulate leaf movement, and how they determine the angle of movement remain to be elucidated. Current studies on leaf movement indicate that there may be a common underlying mechanism of leaf movement in species with nastic movement. In this review, we conducted a detailed overview of how light, endogenous hormones, ions, and aquaporins coordinate to further regulate the regular volume changes within motor organs of the pulvinus and promote leaf movement. This overview focused on *M. truncatula*, *M. pudica*, and *S. saman*, owing to the typical and well-developed motility organs of these plants (Figure 1G–I).

## 2. The Study Species

The species information of *M. pudica*, *S. saman*, and *M. truncatula* was obtained from plant plus of China (https://www.iplant.cn) and accessed on 4 April 2024.

*M. Pudica* is a leguminous, mimosa genus, subshrub herb; the plant can grow up to 1 m in height. The stem is cylindrical and branched, with lanceolate stipules and opposite linear-oblong leaves. *M. pudica* is native to tropical America and widely distributed in tropical regions worldwide, born in the wilderness, shrubs, and perennial shrubs. When touched, the pinnae and leaflets of the mimosa plant close and droop, serving as indicators of changes in weather and rainfall.

*S. saman* is a large thornless tree of the rain tree genus in the family Leguminosae. The crown is extremely spreading and very low-branched, there are often glands between the pinnae and the leaf, and the oblique oblong leaves are gradually smaller from top to bottom. The leaves are the opposite and feathery. The leaves have a circadian rhythm movement phenomenon. At night, they are closed, and they are open in the morning and will be wrapped in dew sprinkled down, like rain. So, it is also called a “rain tree”. Rain trees are tropical and southern subtropical monsoon rainforest species. They prefer light, heat, and humidity, have drought resistance, are barren and coarse, and are one of the fast-growing tree species. Its propagation is generally seeding and cutting propagation.

*M. Sativa* is a legume of the medicago genus of perennial rooted herbs, with stems that are erect, tufted, four-angled, and multi-branched. The stipules are larger and ovate lanceolate, and the leaflets are obovate oblong. The leaves are pinnate ternate compound leaves; there is a circadian rhythm movement phenomenon. The leaves open during the day and are closed at night. It is widely grown in Eurasia and the world for fodder and pasture and is a perennial herb. *M. sativa* has root nodules that provide nitrogen nutrition to the roots, and nitrogen fertilizer is not advocated under general geotechnical conditions.

## 3. Pulvinus: A Representative Motor Organ Composed of Many Cells

The pulvinus is located at the junction of the petiole and the stem (Figure 2A–C), which is elastic and stretchy and can adjust the position of the leaf [1,4,13], and the pulvinus morphology becomes a curved structure during leaf movement. The movement of the pulvinus is mediated by the reversible shrinking and swelling of the flexor cells and extensor cells (Figure 2D–F) [12,13,27]. These two groups of cells, indistinguishable under the microscope, behave in opposite ways and have the same morphological structure [28]. In addition, the cross-section of the *S. saman* pulvinus has a central U-shaped cluster of cells as vascular tissue [28]. Observation of the longitudinal section of the *M. truncatula* pulvinus revealed that the tissue on both sides consisted of irregular polygonal single cells, and the middle semi-circle or fan-shaped was vascular tissue [3,23]. The vascular tissue shape of the pulvinus of *M. pudica* is oval [29]. There was only one central vacuole in the pulvinus cells of *S. saman* and *M. truncatula* [23,30]. However, the pulvinus cells of *M. pudica* contain two types of vacuoles: one tannin rich located near the nucleus and the other aqueous and central, corresponding to the vacuole in most mesophyll cells [31].

The cell structure of both sides of the pulvinus tissues of *M. pudica* had obvious morphological differences [32]. The extensor-side cells were arranged loosely and disordered, while the flexor-side cells were arranged in an orderly and tight manner, containing a large number of secretory cells [29,32]. The epidermal cells of the pulvinus organs of *M. truncatula* are in the shape of folds or sweater patterns [3]. The epidermal cells of the pulvinus organs of *M. pudica* and *S. saman* were smooth without protuberances and contained a large number of trichomes [19,33].

*M. pudica* is known to have a unique seismonasty movement (Appendix A) [34,35]. The exposure of *M. pudica* to a non-damaging stimulus, such as hand touch and wind, generates a weak electric current known as an action potential, which is considered to be the early signal of the leaf’s closing movement [36,37]. Then, the action potential is rapidly transmitted to other tissues by elongated parenchyma cells that are located in the protoxylem and phloem of the vascular bundle and stimulate ion transport in the tissues of the pulvinus to regulate water migration and facilitate leaf movement [35,38]. The parenchyma cells make up the volumetric cortex, which itself is divided into two types of parenchyma cells—flexor cells and extensor cells (Figure 2D) [10,39]. Interestingly, special red cells were found on the adaxial surface of the primer pulvinus of *M. pudica*, and these red cells were demonstrated to be mechanoreceptor cells using anatomies (scanning electron and transmission electron microscopy) and electrophysiological techniques [40]. A similar result was found in the study of the diurnal leaf movement of *Marsilea quadrifolia*, which confirmed that the junction between the blade and petiole was responsible for the reception of light signals and covering the junction with black belt-inhibited leaf movement [1]. In addition, a report pointed out that *Vitis vinefera* (*V. vinefera*) leaves have sharp movement, which is controlled by a unique structure located at the two ends of the petiole, identified as a pulvinus [41] (Figure 2F).

IAA is a critical endogenous hormone that modulates proton extrusion by enhancing the activity of H^+^-ATPase in cell membranes, which drives K^+^ and Cl^−^ fluxes and, consequently, water across the pulvinus into respecifying cells and out of shrinking cells [31] (Figure 3). This causes changes in cell turgor regulated by vacuolar properties, which drives pulvinus movement [11,27]. Notably, there is a similar movement mechanism between the pulvinus’s multicellular movement and the stomata’s unicellular movement, such as being turgor-driven, coupled with bending deformation, and repeatable many times [42,43].

BL is directly sensed by guard-cell phototropin, which induces the phosphorylation of plasma-membrane H^+^-ATPases, the degradation of starch and lipids, and the accumulation of osmolytes within guard cells [44]. Osmosis-driven water uptake increases the volume of guard cells, making them bend and leading to pore opening, whereas the release of osmolyte and subsequent water efflux from guard cells cause the stomata to close. K^+^ and their counter-anions, such as Cl^−^, NO_3_^−^, and malate, are the major osmolytes that regulate the turgor pressure in guard cells [45,46]. The relative contributions of these ions and non-ionic osmolytes, such as sucrose, change drastically depending on the time of the day and environmental conditions [42,47]. Research on the mechanism of stomatal movement has experienced significant progress in recent decades, which provides valuable guidance for future research on pulvinus movement [48].

## 4. Principle of Leaf Movement Based on the Pulvinus Structure

Lacking locomotion to seek out food and safety, plants instead rely on various responses that allow them to fulfill these needs and adapt to changes in their environment [49,50]. Plant responses to external stimuli can be divided into two categories. The first category is rapid movement visible to the naked eye. For example, upon stimulation (e.g., touched by hand), the petiole of the *M. pudica* immediately droops, and the leaf is closed [35]. In the nyctinastic leaf closure of *S. saman* and *M. truncatula*, the leaves naturally unfold during the day and close up at night [12,51,52]. The second category is the micro-movement or slow movement of plant organs. For example, BL receptors are activated by receiving BL, which regulates phototropic movement in *Arabidopsis thaliana* leaves [53] and circumnutations in *Arabidopsis* stems [53,54].

The difference between the two types of movement Is worth noting. In species that show rapid, reversible leaf movement, movement is mediated by turgor changes in specialized cells at the base of the petiole known as the pulvinus or other motor cells [55,56] (Figure 2). In species that lack a pulvinus, like *Arabidopsis*, movement is thought to be due to the differential enlargement of cells in the adaxial and abaxial regions of the petiole [57,58]. Since Darwin published the power of plant movement, scientists have studied it through microscopy, pharmacology, electrophysiology, and histochemical techniques [35,55]. Studies have revealed that plant movement is a complex regulatory network regulated by multiple factors, such as light, hormones, ions, and water transport [11,39].

When the *M. pudica* is exposed to external stimuli, such as hand touch, wind, and insect feeding, it will trigger plants to generate a steep depolarization of the membrane potential known as the action potential [59]. The action potential activates the corresponding ion channel, triggering ion flow and plant movement [60,61]. Precise ion signal transmissions, K^+^, Cl^−^ channels, and aquaporin are thought to regulate the volume of the pulvinus motor cells and, thus, determine leaf movement [18,19] (Figure 3).

Although the mechanism of pulvinus movement has been understood from a biochemical perspective in previous studies, these cellular mechanisms should be supplemented with interdisciplinary research to elucidate the motile mechanism further. A morphological approach has been suggested to understand plant motile mechanisms, such as 2D X-ray micro-imaging techniques and 3D X-ray tomography [4]. In addition, the pulvinus movement can be considered a result of the differential expansive growth of multiple single cells, and “The acid growth theory of auxin-induced cell elongation” to explain the expansive growth of cells has been widely accepted by the plant science community [62,63]. Pietruszka (2021) investigated the problem of reversible interactions between structural elements of the cell wall and established a mathematical model; such modeling could provide noteworthy predictions about the contribution of the so-called acid growth in diverse plant systems, from the single cells level to whole-plant [63]. The “acid growth theory” and “mathematical model” contribute to a deep understanding of a series of challenging scientific questions, such as how ions and water regulate pulvinus movements controlled by multiple cells and how the angle of pulvinus movement is determined.

## 5. Blue Light and Phytohormones Regulate Ion Channels to Promote Pulvinus Movement

As an important external environmental factor, BL acts as a light signal to activate BL receptors, regulate phytohormone levels in tissues, and control plant growth and development and organ movement [14]. Specifically, BL regulates numerous growth and developmental processes during a plant’s life cycle, from seed germination through the early seedling establishment, shade avoidance, establishment of the circadian rhythm, flowering, and even phototropic movement [1,64,65]. Phot proteins (phototropins and homologs) are BL photoreceptors that control mechanical processes like phototropism, chloroplast relocation, or guard-cell opening in plants [66,67]. The changes in the cell-membrane potential of *S. saman* pulvinus flexor-cell protoplasts were detected using the patch-clamp technique during varying illumination, which showed that BL induces the leaflet movement of *S. saman* by inducing K^+^ release from the flexor motor cells [7] (Figure 3). Furthermore, the BL-induced shift of cell-membrane potential was not observed in cells pretreated with a hydrogen-pump inhibitor, suggesting a contribution by the hydrogen pump to the shift [7]. Another study also revealed that a pulse of BL induced both a transient increase in activity of apoplastic K^+^ and membrane depolarization in the laminar pulvinus of *Phaseolus vulgaris* (*P. vulgaris*) [68].

Previous studies have provided substantial evidence confirming that the phototropic bending growth of plant hypocotyls is caused by the asymmetric distribution of auxin in the tissue [5,69,70,71]. A plant’s cell elongation during phototropism movement is considered to be a result of a dynamic balance between water uptake (driven by salt concentration gradients) and cell wall loosening according to the Lockhart equations [72,73]. The lateral transport of auxin in plant cells can be quantitatively described by the physical model established by Pietruszka and Lewicka [73].

It was also demonstrated that the pulvinus movement is likely to be controlled by asymmetric IAA distribution and vacuolar ATPase/PPase activity [15,16,17] (Figure 3). Immunogold labeling of the pulvinus revealed that, in motor cells, those recognizing the A-subunit of the vacuolar H^+^-ATPase were observed preferentially in the tonoplast of the central vacuole [31].

The application of IAA to excise pinna rachises can induce leaf opening in *M. pudica* [15]. Similar results were found for *P. vulgaris* using protoplasts isolated from laminar pulvinus [17]. IAA acts by increasing the H^+^-ATPase activity of the plasma membrane of the cell, decomposing ATP into ADP and inorganic phosphorus and providing energy for the transport of ions through the K^+^ and Cl^−^ channels [21,74,75]. This process enables fast ion transport through plasma membranes.

There is little information about the effect of ABA and GA on the osmoregulation of pulvinus motor cells. Nevertheless, experiments with *P. vulgaris* showed that, contrary to the results obtained with the application of IAA, the application of ABA gives rise to cell shrinking, and its effect is suppressed by an anion-channel inhibitor, suggesting the efflux of Cl^−^ through the activated channels and thus, the charge-balanced efflux of K^+^ [17]. Barley plants and grapes also have similar pulvinus structures [21]. Previous studies have shown that IAA and GA are involved in the movement of the barley pulvinus, with gravity-stimulated pulvinus during inflorescence divided into upper and lower halves, resulting in more IAA in the lower half than in the upper half and higher GA content in the upper half than in the lower half [21]. However, the changes in the two hormones content in both showed a temporal sequence. Interestingly, the expression of *Hv3ox2*, which encodes a key enzyme for the conversion of GA_20_ to GA_1_, was higher on the lower side than on the upper side after 6 h [21]. Furthermore, Yang et al. (2021b) measured the concentration of endogenous hormones in grape leaves and petioles before and after movement, and the results showed that the concentration of IAA in grape leaves and the pulvinus was significantly higher than that in controls but had the opposite effect on GA [76]. After irradiating the abaxial side of treated leaves for 3 h, leaves were sampled for RNA-sequencing analysis, which showed that a total of 44 genes were involved in the hormone signaling pathways of IAA, BR, ABA, GA, ethylene (ETH), CTK, jasmonic acid (JA), and salicylic acid (SA), including 32 up-regulated and 12 down-regulated genes [76].

This series of studies indicated that leaf movement is driven by the gradient distribution of auxin in the tissues on both sides of the pulvinus; that is, the content of IAA in the extensor tissue is higher than that in the flexor tissue (Figure 3). In addition, one of the most critical regulatory small molecules in plants is IAA, which is primarily synthesized from tryptophan through the TAA1-YUC pathway [67,77] (Figure 3). The conservation of this pathway in the plant kingdom has been functionally validated in many plant species. Five TAA and 11 YUC gene members were identified in the *Arabidopsis* genome [78,79]. TAA proteins appear to be broadly and stably expressed in all organs, whereas YUC enzymes show more pronounced organ-specific expression [80]. *BnaYUCCA6*, a homolog of *Arabidopsis* YUCCA6 in the oilseed rape (*Brassica napus* L.), was identified as affecting leaf angle regulation [81]. YUC2, YUC5, YUC8, and YUC9 are essential for induced hypocotyl bending growth and phototropism in *Arabidopsis* [82]. YUC2, YUC5, YUC6, YUC8, and YUC9 may also be involved in regulating pulvinus movement. In addition, the study on how TAA and YUCCA members jointly regulate leaf movement and their specific expression in the leaf and pulvinus has not been reported, so far. The above description of YUCCA-related studies is only due to its regulation of similar morphogenesis (leaf angle, hypocotyl bending) with pulvinus movement morphological changes, which can be used as a further reference direction for further research. 

IAA dynamic redistribution has a crucial role in almost every aspect of plant life, ranging from cell shape or volume change and division to organogenesis and responses to light and gravity [83]. However, owing to technical limitations, little is known about the actual distribution of auxin in tissues at single-cell resolution. Currently, plant biologists can only use markers to visualize growth-hormone distribution, such as the expression of growth auxin-dependent reporter genes (e.g., using the systems DR5::GUS3; DR5: ER-GFP; and DR5: NLS-3xGFP) [84]. A limitation of these methods is their irreversibility, which precludes visualization of transient changes in auxin levels. Fortunately, a study showed a genetically encoded biosensor for the quantitative in vivo visualization of auxin distribution [84]. This sensor enables direct monitoring of auxin’s rapid uptake and the clearance of auxin by individual cells and within cell compartments in the plant [84]. Altogether, BL and phytohormones influence K^+^ channels to trigger leaf movement, and the uneven distribution of auxin in the pulvinus tissue is the main factor of leaf movement. However, the regulatory effects of other hormones need more scientific experiments.

## 6. Dynamic Balance of K^+^ and Cl^−^ Ions Determines the Pulvinus Osmotic Potential

As the motor organ of *M. truncatula, M. pudica*, and *S. saman* (Figure 1G–I), the ability of the pulvinus to control the movement of the leaf and petiole is regulated by many factors. The dynamic migration of K^+^ and Cl^−^ governs the changes of cell turgor and promotes plant organ movement [11]. Multicellular pulvinus movement and stomatal guard-cell morphological changes may exist in similar regulatory mechanisms [6]. In the vacuole, K^+^ ions and their counter anions, such as CI^−^, NO_3_^−^, and malate, are the main osmolytes that regulate guard-cell swelling pressure [46]. Closure of the stomata involves the efflux of anions, predominantly via SLOW ANION CHANNEL 1 (SLAC1)-type channels from the SLACI/SLACI HOMOLOG (SLAH) family [46]. Ion transport regulates the acidity of the vacuole, and osmotically driven water uptake increases the turgor pressure of guard cells, causing them to bend and leading to stomatal opening [6].

K^+^ is an essential monovalent cation for plant growth and development and a key ion for regulating osmotic balance in cells [19,42]. Using energy-dispersive X-ray microanalysis, the content of K^+^ and Cl^−^ in the primary pulvinus tissues of *M. pudica* were analyzed, and the results showed that the concentration of K^+^ in the apoplast of external extensor cells decreased significantly, while changes in the Cl^−^ content were less striking in the pulvinus [6,85]. The accumulation of ions in the apoplast seems to be initiated by the decrease in water potential triggered by an apoplastic accumulation of unloaded sucrose [32]. Likewise, Toriyama revealed that K^+^ migrated from the intracellular to the extracellular space of the motor cells after movement, indicating that water migration was osmotically driven by K^+^ migration, resulting in the loss of turgor of the pulvinus of *M. pudica* [86]. This conclusion was confirmed by Allen using a ^42^K^+^ efflux experiment, and it was also reported that the Cl^−^ concentration of flexor cells of the pulvinus was significantly higher than that of the extensor cells after stimulation, indicating that the dynamic balance of K^+^ and Cl^−^ are required for pulvinus movement [59]. In addition, following stimulation of the pulvinus, unknown colloidal substances migrate from the central vacuole into the extracellular space of the pulvinus motor cells, suggesting that these substances diffuse from the motor cells at the same time as K^+^ migration [86]. In addition, Samejima and Sibaoka (1980) suggested that the efflux of fluid from the central vacuole reduces the volume of the motor cell, resulting in rapid bending of the pulvinus. These studies suggest that efflux of vacuolar material from motor cells may directly regulate the volume of motor cells [87].

In addition, patch-clamp studies revealed that, in the whole-cell configuration, K^+^-influx currents were observed in extensor cells from the pulvinus of *S. saman* [88]. For the dark-promoted leaf closure of *S. saman*, the involvement of K^+^ fluxes, driving the hydraulic movement of water, was unequivocally also demonstrated using motor-cell protoplasts [18]. Furthermore, the specific K^+^ channel blocker, TEA, completely inhibited the shrinking of extensor-cell protoplasts, pointing to the involvement of potassium in induced protoplast shrinking, resulting in the leaf movement of *S. saman* [51]. The ion uptake appears to be coupled to the H^+^ gradient produced by the activity of H^+^-ATPases [75] (Figure 3).

It is also possible that motor cells absorb apoplastic K^+^ through inward-rectifier K^+^ channels, which are activated by H^+^ pump-driven plasma-membrane hyperpolarization [28] or other mechanisms [12]. It was later confirmed that K^+^ was transferred via the SPICK2 K^+^-influx channel, SPICK2, a homolog of the weakly inward-rectifying Shaker-like *Arabidopsis* K channel, AKT2, and is regulated in vivo directly by phosphorylation [89]. Nieves et al. reported a new grapevine Shaker-type K^+^ channel, *VvK3.1*, and revealed that *VvK3.1* belongs to the AKT2 channel phylogenetic branch and is a weakly rectifying channel, mediating both inward and outward K^+^ currents [41]. Moreover, the high amount of *VvK3.1* transcripts detected in the pulvinus strongly suggests a role for this Shaker in the swelling and shrinking of motor cells involved in leaf movement [41].

In addition, K^+^ channel activity in the net steady-state outward K currents was inversely related to the plasma-membrane phosphatidylinositol (4,5) bisphosphate (PtdInsP_2_) levels, short-term manipulations decreasing PtdInsP_2_, increased NtORK (outward-rectifying K^+^ channel) activity, increasing PtdInsP_2_, and decreasing NtORK activity [90] (Figure 3). Despite these, studies can demonstrate that potassium ions are the key to pulvinus movement; the potential coordinating transport mechanisms of potassium ions and other ions need to be further clarified.

## 7. The Dynamic Changes of Actin and Ca^2+^ Regulate Pulvinus Movement

Transient and spatial dynamic changes in Ca^2+^ in plant tissues are critical for plant growth and development, signal transduction, stomatal movement, salt or osmotic stress response, the defensive behavior of plants, and circadian rhythm movement of leaves and pulvinus [11,91,92].

In previous research, circadian rhythm movement or seismonasty movement in legumes (e.g., *M. pudica*, and *S. saman*) has been described to be due to the active transport of Ca^2+^ ions in the intracellular and extracellular [11,35]. Scientists have used *M. pudica* as a model organism for researching leaf or pulvinus movement because of its fast-moving feature. Interestingly, compared with other model plants used to study leaf movement, *M. pudica* contains two types of vacuoles in the motor organ [3,93], including the tannin and central vacuoles. Tannin vacuoles contain great amounts of tannin, are located near the nucleus, and are supposed to function as potential Ca^2+^ stores [94]. The central vacuole does not contain tannin, is much larger than the tannin vacuoles, and occupies a central position in cells [27]. During shrinkage, both vacuoles change their shape [27]. Quantitative data obtained by microspectrophotometry demonstrated calcium migration during the bending movement of the primary pulvinus [95]. In the extensor motor cell, a small amount of calcium migrates from the tannin vacuole, and calcium on the cell wall moves to the central vacuole. In the flexor cell, a large amount of calcium from the tannin vacuole moves to the central vacuole of the motor cell [95]. The recovery and maintenance of the tannin vacuole may play a role in maintaining turgor in the motor cells of the flexor of the primary pulvinus of *Mimosa* [95]. The central vacuole is the intracellular store of solutes and water, suggesting that during contraction or swelling of a motor cell, the vacuole releases or accumulates ions through channels and transporters, as in stomatal movements [35].

The actin cytoskeleton consists of two forms of actin, globular actin (G-actin) and filamentous actin (F-actin) [96]. The two forms of actin are dynamically converted and regulated by a plethora of actin-binding proteins (ABPs) [97,98]. This dynamic conversion and regulation are essential for various plant physiological processes [99]. The functions of Ca^2+^ and actin were described in more detail in another review [98]. The pulvinus of *M. pudica* was treated with actin-affecting reagents and calcium channel inhibitors, which confirmed that actin dynamics mediate the calcium level changes during the pulvinus movement of *M. pudica* [100,101] (Figure 3).

Although the contribution of Ca^2+^ and actin to pulvinus movement has been established, the regulatory relationship between them is unclear [20,34,95,102]. Toriyama et al. (1970) used the microscopic localization of Ca^2+^ in the motor cell, and pulvinus obtained before or after stimulation were fixed in Lillie’s neutral buffered formalin [95]. They found that the effluent from stimulated pulvinus has significantly more Ca^2+^ than that from unstimulated controls [95]. Additionally, there is evidence that a low concentration of EDTA (1 mM) retards the reopening process, while a higher EDTA (10 mM) concentration prevents the closing movement of the rapid leaf movements of *M. pudica* [102]. In contrast, the rapid restoration of normal closure is attainable by transferring the samples to CaCl_2_, suggesting that EDTA removes Ca^2+^ from a readily accessible site and causes easily reversible, rather than profoundly disruptive, effects [102]. A study by Raeini-Sarjaz et al. (2011) on the diurnal leaf movement of *P. Vulgaris* yielded similar results and further confirmed that calcium ions are indispensable in leaf movement [22]. Kameyama et al. (2000) found that actin filaments in the motor cells at the lower side of the pulvinus, but not at the upper side, become more peripheral after bending. Therefore, the bending of *M. pudica* is correlated with reduced actin phosphorylation in the pulvinus, indicating that actin phosphorylation is essential for pulvinus bending resulting from actin cytoskeletal alterations [34,103]. A similar interaction between actin dynamics and Ca^2+^ influx was observed on the plasma membranes of stomatal guard cells [104].

In addition, research on the labellum closure movement of the *Venus flytrap* (*V. flytrap*), a type of plant with distinct motor characteristics but no typical motor organs, further confirmed that Ca^2+^ plays an indispensable role in regulating the movement of plant organs [93]. Based on the aforementioned evidence, we can speculate that the interaction between actin and Ca^2+^ in the pulvinus tissue is indispensable for the pulvinus.

## 8. Aquaporins Mediate Water Transport to Facilitate Pulvinus Movement

Aquaporins (AQPs) are universal membrane channel proteins that selectively and reversibly promote water movement across plasmalemma and organelle membranes in plants and other organisms, resulting in the expansion or contraction of cell volume to generate driving force and promote the movement of plant tissues and organs [38,105]. They belong to the membrane-integrated major intrinsic super-family proteins (MIPs) with small molecule weights (about 26 to 34 kDa), and numerous members have been found in all kingdoms from Archaea to animals [38,106]. Based on their sub-cellular locations and sequence similarities, and compared with the mammalian aquaporins, plant aquaporins can be further divided into five major groups [38,107]: the plasma-membrane intrinsic proteins (PIPs); tonoplast intrinsic proteins (TIPs); nodulin 26-like intrinsic proteins (NIPs), (NIPs: where NOD-26 is an aquaporin discovered in the peribacteroid membrane of nodulated soybean roots); small basic intrinsic proteins (SIPs); and uncategorized intrinsic proteins (XIPs) [38,106]. PIPs and TIPs are dominant and mainly mediate water flow across cells and sub-cellular compartments in higher plants. PIPs can be further divided into PIP1 and PIP2 subgroups [108]. For the fine classification and function of other aquaporins, please refer to the article written by Wang [38].

PIPs represent the most abundant aquaporins in the plant plasma membrane and display observably cell-specific expression patterns in the leaves or pulvinus of several movement model plants [18,103,109]. Using patch-clamp functional analysis in *Xenopus oocytes* showed that the production of PIP2 aquaporins increases the water permeability of the oocyte membrane by about 10- to 20-fold [110]. In contrast, PIP1 aquaporins alone do not evidently affect the membrane’s water permeability [18,111]. The rapid movement of water through the biomembrane is crucial for the maintenance of cellular water homeostasis and for the accomplishment of various metabolic activities under many circumstances, such as during cell elongation, root water absorption, stomatal opening and closure, flowering, fertilization, and, especially, leaf movement [106,107,112]. Hence, aquaporins are greatly essential for plant growth, development, and adaptation to changing environments. The leaf movement of *M. pudica* is inseparable from aquaporins-facilitated water uptake and transport in the pulvinus [11,31,33].

Using a nuclear magnetic resonance (NMR) imaging technique revealed that, after stimulation of an *M. pudica* plant, water in the flexor half of the primary pulvinus disappeared; the water previously contained in this area seemed to be transferred to the upper half of the primary pulvinus [33]. During the pulvinus recovery phase, water was transferred from the upper part to the lower part [33] (Figure 3). The hypothesis that water flows from the flexor side to the extensor side needs more evidence because the possibility of water transport on the same side cannot be ruled out. In addition, the abundance of aquaporins in the vacuolar membrane of the pulvinus of mature motor cells of *M. pudica* increased by more than 3-fold compared with the juvenile pulvinus [75]. cDNAs encoding aquaporins, PIP1;1 and PIP2;1, were isolated from *M. pudica* seedlings’ cDNA library, and in *Xenopus oocyte,* a function co-expression analysis showed that MpPIP1;1 itself displayed no water channel activity. It facilitated the water channel activity of MpPIP2;1 in a phosphorylation-dependent manner [103]. Further immunoprecipitation analysis revealed that phosphorylation of Ser-131 of *MpPIP1;1* binds directly with *MpPIP2;1* and is involved in regulating the structure of the channel complex and affecting water channel activity [103]. Moreover, in the motor cells of *S. saman*, two plasma membrane intrinsic protein homolog genes, *SsAQP1* and *SsAQP2*, were cloned from these organs and characterized as aquaporins in Xenopus laevis oocytes. The aquaporin mRNA levels differed in their spatial distribution in the leaf and were regulated diurnally in phase with leaflet movement [18]. Additionally, SsAQP2 transcription was under circadian control, and these results link AQP2 to the physiological function of circadian rhythmic cell movement [18].

The above results have revealed a hypothesis that AQP1 and AQP2 bind to form a complex and interact with each other to enhance the abundance of AQP2, further regulating water migration of the pulvinus extensor and flexor cells, thereby facilitating pulvinus unite leaf movement.

## 9. Molecular Regulatory Network of *Medicago truncatula* Leaf Movement

*M. truncatula* is a model plant for studying legumes, with two typical features found in its leaves, including diverse compound forms and the pulvinus-driven nyctinastic movement [52] (Figure 4). In the last decade, scientists have carried out numerous studies on *M. truncatula* petal symmetry shape, leaf compound morphogenesis, motor organs, plant height, plant branching through mutant screening, and functional analysis strategies [3,23,24,25,26,113,114,115]. However, the underlying molecular mechanism of the nyctinastic leaf movement of *M. truncatula* is just beginning to be uncovered, providing new insight for further exploring leaf movement [23,52].

Using a genetic approach, Chen et al. isolated a mutant designated elongated petiolule1 (elp1) from *M. truncatula* that fails to fold its leaflets in the dark due to loss of motor organs. Map-based cloning indicated that ELP1 encodes a putative plant-specific LOB domain transcription factor, which is expressed in primordial cells that give rise to the motor organ [23]. Identifying ELP1 orthologs from other legume species showed that motor organ identity is regulated by a conserved molecular mechanism [23]. Furthermore, they found that two allelic mutants of *M. truncatula* with unclosed leaflets at night were impaired in *MtDWARF4A* (MtDWF4A), a gene encoding a cytochrome P450 protein orthologous to *Arabidopsis* DWARF4 [52]. Surgical experiments and genetic function analysis of double mutants revealed that the geometry of the compound leaf plays a significant role in the leaf movement of *M. truncatula* modulated by MtDWARF4A [52]. In addition, Zhou et al. (2021) revealed that MINI ORGAN1 (MIO1), a gene encoding an F-box protein SMALL LEAF AND BUSHY1 (SLB1), not only regulates the organ size of *M. truncatula* but also controls the development of pulvinus and leaf movement [52]. The mutant functional analysis also showed that the nodulation and nyctinastic leaf movement are controlled by the clock gene *MtLHY* [116] (Figure 4). These results imply that a common signal pathway may regulate organ development, leaf morphogenesis, and leaf nyctinastic movement. However, the above speculations must be further justified.

## 10. Conclusions and Perspectives

In this review, we have described the biomechanical and molecular aspects of pulvinus-driven leaf movements. Over the past decades, much has been learned about single ion function using special ion detection instruments that can only measure the final ion accumulation value or by using simplified experiment environments that treat with single exogenous drugs, hormones, and light. However, how these ions coordinate their dynamic transport with each other remains unclear. Much remains to be understood about ion transportation or hormonal regulation at the tissue and cellular levels. The Figure 3 model only reflects the movement mechanism of the pulvinus when the leaf is closed but does not represent the movement mechanism of the pulvinus when the leaf is opening. The mechanism of pulvinus movement during leaf opening is the main direction of future research.

In addition, the current information nowadays enables researchers to explore more realistic natural circumstances to investigate the mechanisms underlying the integration of signals deriving from different ions, different hormones, and a complex light environment [13,84,93,117]. For example, a genetically encoded biosensor for the quantitative in vivo visualization of auxin distribution enables direct monitoring of auxin’s rapid uptake and clearance by individual cells and within cell compartments in plants [84]. Recent advances in transgenic technologies [42,93,118] and whole-genome sequencing of fast-moving plants [119], as well as the rapid development of genome editing technologies [120], will lead to technological breakthroughs in this compelling and challenging area of research.

## Figures and Tables

**Figure 1 ijms-25-04582-f001:**
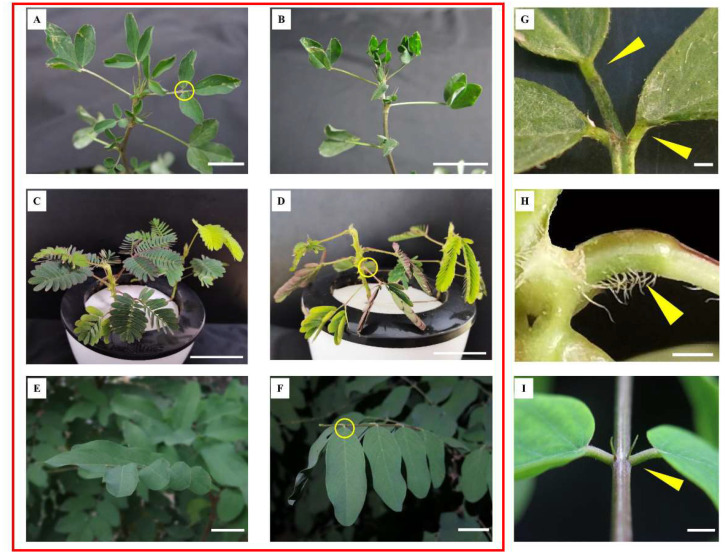
Leaf movement of *Medicago truncatula*, *Mimosa pudica*, and *Samanea saman.* (**A**,**B**) represent the images of *M. truncatula* leaf opening in the daytime and the leaf closing at night, respectively. Scale bars, 20 mm. (**C**) represents the open state of *M. pudica* leaves in the daytime or without external stimulation. Scale bars, 20 mm. (**D**) represent the closed state of *M. pudica* when it is stimulated by external stimulation. Scale bars, 20 mm. (**E**,**F**) represent the images of *S. saman* leaf opening in the daytime and leaf closing at night, respectively. Scale bars, 20 mm. (**G**) Close-up of the motor organs of *M. truncatula*; The yellow triangle points to the pulvinus. Scale bars, 4 mm. (**H**) Close-up of the motor organs of *M. pudica*; the yellow triangle points to the pulvinus. Scale bars, 4 mm. (**I**) Close-up of the motor organs of *S. saman*; the yellow triangle points to the pulvinus. Scale bars, 4 mm. The yellow circle is where the motor organs are located. Generally, the leaf is open from 7:30 a.m. to 8:00 p.m. and closed at night.

**Figure 2 ijms-25-04582-f002:**
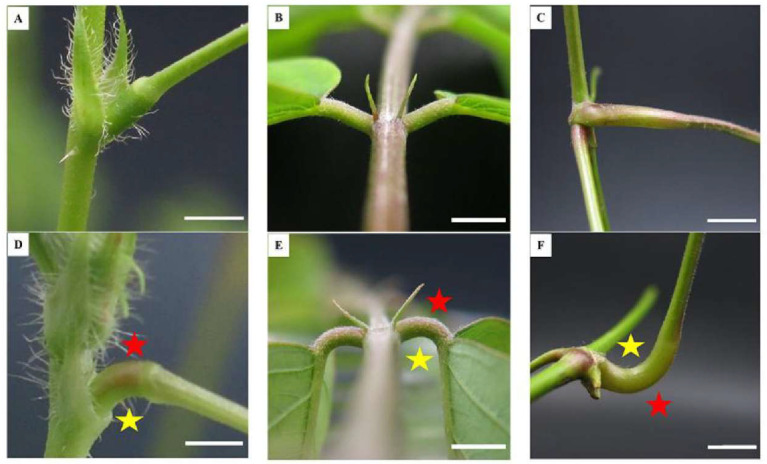
Motor organs of *Mimosa pudica*, *Samanea saman*, and *Vitis vinefera.* (**A**) is the unbent primary pulvinus of *M. pudica*. Scale bars, 4 mm. (**B**) is the unbent pulvinus of *S. saman*. Scale bars, 4 mm. (**C**) is the unbent pulvinus of *V. vinefera*. Scale bars, 4 mm. (**D**) is the bent primary pulvinus of *M. pudica* after external stimulation or circadian rhythm leaf movement. Scale bars, 4 mm. (**E**) is the bent pulvinus of *S. saman* after circadian rhythm leaf movement. Scale bars, 4 mm. (**F**) is the bent pulvinus of *V. vinefera*. Scale bars, 10 mm. The front side of grape leaves is fixed towards the ground; about 5 h later, the leaf front faces up and the pulvinus is obviously bent. The red pentacle represents extensor cells on one side, the yellow pentacle represents flexor cells on the other.

**Figure 3 ijms-25-04582-f003:**
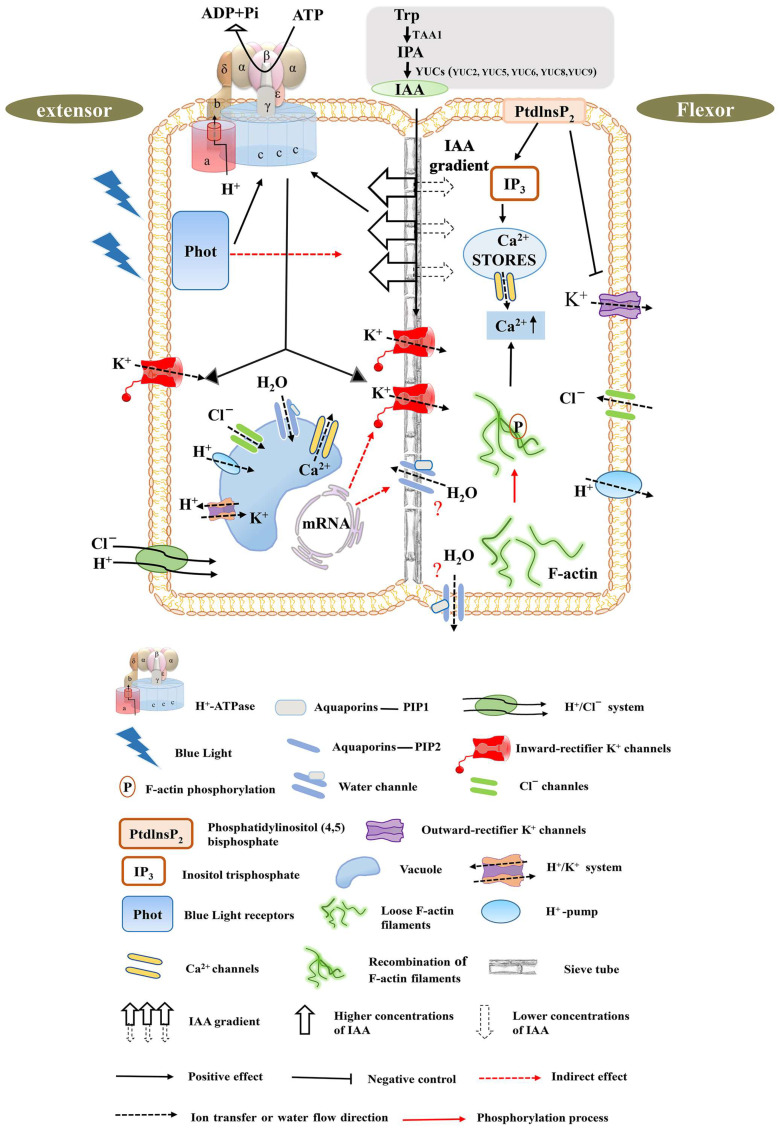
Movement mechanism of the pulvinus motor organ—a model. This model mainly represents the movement mechanism of the pulvinus when the leaf is closed, but does not represent the movement mechanism when the leaf is open. Left: schematic diagram of extensor cells swelling during pulvinus movement. Right: schematic diagram of flexor cell shrinking during pulvinus movement. Below is the key for symbols and abbreviations. Prior to osmotic transport, a blue-light receiver, Phot, senses changes in light, which generates activation of TRP, TAA1, IPA, and YUS auxin synthesis pathway elements. Differential gradient auxin transport in flexor and extensor sides serves as an early signal of cell volume change. Outward/inward-rectifier K^+^ channels are activated by blue-light receptors, which regulate the distribution of K^+^ in tissues. Aquaporins of the plasma-membrane intrinsic protein (PIP) family increase the permeability of biological membranes to water and facilitate the transverse and longitudinal transport of water on the extensor and flexor tissue sides. The actin cytoskeleton consists of two forms of actin: globular actin (G-actin) and filamentous actin (F-actin), where F-actin transitions from a loose state to a reorganized ordered phosphorylation state that regulates calcium ion levels in the vacuole or cytoplasm. Calcium ion levels are also regulated by phosphorylation of phosphatidylinositol (4,5) diphosphate. In the pulvinus, H^+^-ATPase activity exists with a difference, whereas the activity of the extensor side was higher than that of the flexor side. The dynamic transport of cation (H^+^) and anion (Cl^−^) regulates the charge balance, and the differential transport forms an electrochemical potential gradient to provide energy. The differential transport regulation of hormones, ions, and water results in higher turgor pressure in the extensor than the flexor, forming a certain pressure difference and providing a driving force for the movement of the pulvinus. Notes: Cellular water content as well as cytoplasmic pH are very stable. Cells with protein storage vacuoles and cells with solubilizing vacuoles have very similar water contents and cytoplasmic volumes, but vacuole volume is different.

**Figure 4 ijms-25-04582-f004:**
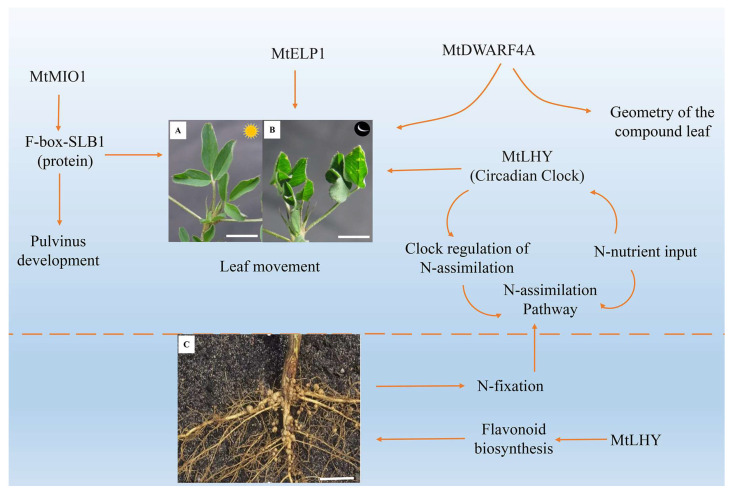
Molecular regulation mechanism of nastic movement of *Medicago truncatula* leaves. (**A**,**B**) represent the unfolding state of *M. truncatula* leaves during the day and the closing state at night, respectively. Scale bars, 20 mm. *MtMIO1* encodes F-box protein SLB1 regulates the development of the pulvinus organs and controls the circadian movement of leaves; mutations in the *MtELP1* gene lead to the loss of pulvinus, preventing the leaves from closing at night. MtDWARF4A can regulate leaf movement and the geometry of the compound leaf. (**C**) represents the root structure of *M. truncatula*, Scale bars. 100 mm. *MtLHY* controls the nodulation and nyctinastic leaf movement. The yellow arrow indicates the regulatory function.

## Data Availability

No new data were generated or analyzed in this study.

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
