# Peer review of "Mechanism of the Pulvinus-Driven Leaf Movement: An Overview"

_ijms, 2024, doi:10.3390/ijms25094582_

Round 1

Reviewer 1 Report

Comments and Suggestions for Authors

This review on the mechanism of pulvins on plants movement is well written and organised.

The abstract is a reflect of the main elements known about plants movement. The figures present are well described and really clear.

The introduction provides a general view on the movement in plant, based on the three chosen model plants. Then they describe how pulvins are structured and the external factors that make plants move.

Then they describe the how the pulvins evolve to conduce to the plants movement, and the implication of hormons and blue light. Then the different channels of K+, Cl- and Ca+ and aquaporins implications are described. To the end they start to identify the molecular regularions with one example.

The conclusions are online with the documented data and open to new potential researches.

Just need to be checked the beginnng of line 115.

Reviewer 2 Report

Comments and Suggestions for Authors

The paper devoted to review investigations of the regulation of leaf movement in response to envinronmental signal.

Authors focused on several model plants, Medicago, Mimosa and  Samonea.

However, model, proposed by authors (fig 3) ignore role of vacuole in turgor and cell volume regulations. This points need to be added and discussed in details.

Some more specific comments.

Abtracts:

Require significant corrections.

For example, lines 10- 13: very complicated and confuse sentence. Authors mixed primary and secondary signaling etc. GA is also a growth hormone (phytohormone), indeed. Many other senteces in the abstracts have the similar rpoblem. Authors need to adjust the punctuation to make text clear and readable.

Line 25: „adaptation by plants to adapt” ¿?

Lines 57 – 62: very confused part. What do authors means as IAA? IAA (auxin) is main hormone in plants produced in different location, somethimes assymetrically and asymetricaly distributed. This is the meain reason of changes in organ orientation. Exogenous treatments with IAA or other hormones may not serve as good model for study organ orienation in the space (movement)!

Line 63: „ The accurate ion signal” is not a very scientific point, indeed!

Line 65: “the volume of moving cells”?? please, consider that the volume of cells regulated mainly by vacuolar volume, ea. Vacuolar transporter and vacuolar pH. Acidification of vacuole linked with cell expansion.

Lines 72 – 75: very complicated statement, not scientifically precise.

Lines 80 – 82: why twice words “leaf movement“ ???

Lines 110- 133: it will be nice to describe in more details differences in cell structures.

Lines 137:“ KCl fluxes“ ??

Line 139: turgor regulated by vacuolar properties, indeed.

Figure 3 seems to be a very problematic. Plant cell volume  and turgor pressure regulated by water contents in vacuole, not in cytoplasm. And tonoplats enzymes serve as a main player in this  process. However, on figure 3 vacuaole even did not mention.

Moreover, as upstream signal, authors shown IPA-YUC_IAA pathway. However, this need to be specified: plant have numbers of YUCCA genes, located in different places and responsible for different processes: cell division in meristem and stem cell maintenace (IPA), auxin production and canalization to the root (YUCCA4 in arabidopsis) etc. The similar occurred in the other species.

Plesae, be very precise and re-draw model.

Line 258: cell wall expansion (loosening???) is not enough and can not serve as driving force of cell growth (change volume) and organ movement. Cell elongation/growth occurred by vacuole growth , extension of plasma membrane and vacuolar membrane by supplying of new phospholipids.

These points are missing in the current manuscript. 

Lines 261- 262: not by IAA, but by assymetric IAA distribution and vacuolar ATPase/PPase activity.

Line 307: it will be nice to add information that Cl mainly accumulated in the vacuole, acidify it and increase turgor pressure through this mechanisms.

Line 469: Medicago, not medicago.

Lines 480 – 496: why did you change font size?

Fig 4: scale bar?

Comments on the Quality of English Language

Significant corrections and more clarity are required!

Reviewer 3 Report

Comments and Suggestions for Authors

Dear Authors,

The submited manuscript titled „Mechanism of the pulvinus-driven leaf movement: an overview” is generally well-planed and well-written and contains very interesting results, which might interest an international audience. Nevertheless, I have found some imperfections, which (in my opinion) should be corrected or at least clarified before an eventual publication. I have listed them below:

1.       Line 84-85 I suggest to add at least short description of studied species (range, habitat affiliation, lifespan etc.) perhaps in the chapter titled „The study species”.

2.       I think, that very valuable would be add section of Material and methods containing at least brief description of literature collection mode (if the databases such as WoS, Scopus etc. were used? If so which key words were applied? What criteria of inclusion/exclusion of literature sources were applied?). Such section will justify the choice of publication included into owerview.

3.       Figue 3 is a bit illegibile, please enlarge it.

4.       The list of References as well as mode of incorporation of literature sources in the text should be prepared according to guidelines for Authors.

Round 2

Reviewer 2 Report

Comments and Suggestions for Authors

Thank you for great work! It is better now. However, there are some points still unclear.

Line 311-318: TAA1 and YUCCA expressed not in all organs, but in specific cell type. This is a key point in leaf movement. Please, clarify this point.

Lines 321 - 322: However, it has not been possible to directly determine IAA spatial and temporal distribution at a cellular resolution" - this is not true. There are a lot of markers.

Figure 3 is OK now. Plesae, also, consider that water content as well as cytoplasmic pH is very stable. In the cell with PSV (proten storage vaciole) and in the cell with LV (lytic vacuole) realktove water contents as well as cytoplasmic volume is very close, but vacuole volume is different.

Please, slightly clarify this point.

Comments on the Quality of English Language

minor editing during proof-reading
